# Effectiveness of a Social-Media-Based Diet and Physical Activity Programme for Fathers in Japan: A Randomised Controlled Trial

**DOI:** 10.3390/ijerph21081104

**Published:** 2024-08-21

**Authors:** Kayo Maruyama, Kumiko Morita

**Affiliations:** Graduate School of Health Care Sciences, Public Health Nursing, Tokyo Medical and Dental University (TMDU), 1-5-45 Yushima Bunkyo-ku, Tokyo 113-8519, Japan; morita.phn@tmd.ac.jp

**Keywords:** social media, lifestyle, health education, diet, physical activity, fathers, children, intervention, randomised controlled trial

## Abstract

Provision of healthy lifestyle support for fathers can improve the lifestyles and health awareness of not only fathers but also their children and families. Few studies have examined the effectiveness of education targeting healthy fathers provided via social media. Hence, we aimed to clarify the effects of providing fathers with information on healthy lifestyle habits via social media on their awareness and behaviours regarding diet, physical activity, and healthcare, in addition to such a programme’s indirect effects on their children. This randomised controlled trial included 73 fathers of primary school children in Japan. The intervention group received information on ‘healthy lifestyle’, ‘lifestyle-related diseases’, ‘healthy diet’, and ‘healthy physical activity’ via social media six times every 2 weeks. Data were collected before, 3 months after, and 4 months after the intervention. The intervention was effective in improving some awareness regarding diet, physical activity, and healthcare. In addition, the fathers in the intervention group demonstrated significantly improved interest in their child’s diet and exercise habits. Social-media-based diet and physical activity programmes for fathers improved their awareness and behaviour. Therefore, social-media-based health education programmes can be an important tool for increasing fathers’ interest in their own health and in their children’s lifestyles.

## 1. Introduction

The World Health Organization defines non-communicable diseases (NCDs) as diseases caused by unhealthy lifestyles (e.g., an unhealthy diet, physical inactivity, exposure to tobacco smoke, and excessive alcohol consumption) [1]. NCDs include cancer, cardiovascular disease, and chronic respiratory disease and currently account for 60% of deaths in Japan [2]. The metabolic risk factors that heighten the risk of NCDs increase after 40 years of age [3]. To prevent the development of NCDs, people should adopt healthy lifestyles and appropriate habits from an early age, such as childhood or school age. Lifestyle habits acquired during childhood are maintained over the long term [4] and have a major impact on health throughout life. Children’s lifestyles and health are related to their family environments [5]. Therefore, it is considered important to involve parents in health education for children to help them develop healthy lifestyle habits [6,7]. Several RCTs have shown that intervention studies involving parents are effective in improving the lifestyles of both children and parents [8,9,10,11,12]. Traditionally, mothers are considered the primary caregivers of children [13]. Several studies have reported a distinct association between mothers and children’s lifestyle habits [14] and underestimated the influence of fathers [15]. However, recent studies focusing on fathers have shown that changes in fathers’ behaviour can influence the lifestyles of their children [16,17,18,19]. Accordingly, accumulating evidence is showing the influence of fathers on their children’s behaviour, which is strongly associated with children’s dietary intake as well as fathers’ eating habits [20]. In terms of exercise habits, fathers who engage in sustained physical activity exert a greater influence on their children’s physical activity than mothers [21]. In Japan, fathers’ lifestyle and involvement with their children may positively influence their children’s health and development [22,23]. In recent years, with the increase in the number of nuclear families and dual-earner households in Japan [24], the roles expected of fathers have changed, and there is a renewed awareness of the importance of a father’s presence in their child’s development process, as he is responsible for childcare and housework at home in the same way as the child’s mother.

To date, research on fathers helping their children to adopt healthy lifestyles has been limited to fathers with obesity [16,17], and few studies have focused on healthy fathers [18,19]. In terms of methods, face-to-face interventions are predominantly used; meanwhile, only a few studies have examined online social media interventions. According to data from the Organisation for Economic Co-operation and Development on international comparisons of lifetimes, Japanese men have a great deal of paid-work time (442 min/day) [25], and fathers with >12 h of work-related time spend limited time on housework and childcare (average 10 min/day) [26]. Social-media-based health education is a convenient and innovative method for Japanese fathers raising children under such circumstances. The internet usage rate in Japan is reported to be over 80%, with smartphones having the highest internet usage rate by device, 70% [27]. The programmes proposed in this study were principally available via smartphone applications; therefore, the participants could follow them at home on their own time and at their own pace [28,29]. This advantage may have helped the target group to participate in the programme. With this vast ability to reach the general public, social media has had a significant social impact in Japan [30] and will be increasingly used in the field of health education in the future. In addition, the ability to obtain correct health information from experts through apps is an effective way of fostering health awareness. It is expected that fathers will be able to reflect with their children on their past lifestyles and develop appropriate lifestyles during their children’s school years, when the foundations of their lifestyles are being formed, thereby improving their own and their children’s lifelong health.

Therefore, we aimed to clarify the effects of providing fathers with information on healthy lifestyle habits via social media on their awareness and behaviours regarding diet, physical activity, and healthcare, in addition to such a programme’s indirect effects on their children. We hypothesised that fathers in the intervention group would demonstrate improvements in awareness and behaviours in relation to their diets, physical activity, and healthcare, as well as their children’s lifestyle habits, brought about by the programme. The programme commenced on 3 May 2022 and ended on 10 December.

## 2. Materials and Methods

### 2.1. Study Design

This trial was registered with the Japan Registry of Clinical Trials (jRCT).

The registration number is jRCT1030210473 (https://jrct.niph.go.jp/, accessed on 14 July 2023). The first registration occurred on 10 December 2021. This randomised controlled trial evaluated the effectiveness of a programme in supporting the healthy lifestyles of fathers of primary school children. In this prospective study, the participants were randomly assigned to intervention and control groups (waitlists) in a 1:1 ratio. Participant allocation was carried out in random order using a computer-based random number generation algorithm. To ensure equal allocation to each group, a random number was generated for half of the participants, with a block size of 2. The participants were allocated to either the intervention or control group, with block sizes of >0.5 or <0.5, respectively. The participants, care providers, and those assessing the outcomes were not blinded to the allocation groups.

### 2.2. Participants

This study included fathers with children aged 6–12 who owned a smartphone and had access to an app specified by the researchers. Meanwhile, participants who at the time were under dietary and exercise restrictions at the behest of their healthcare providers were excluded. There were two reasons for targeting fathers of school-age children in this study. First, school-aged children are in their formative years regarding their lifestyles, an important period that has a significant impact on their lifestyles in adulthood [31]; second, for fathers, this is the age when they are more susceptible to lifestyle-related diseases [3]. For these reasons, we thought it would be worthwhile for both children and fathers to reflect on their lifestyle habits during this period. The participants were recruited through Facebook, Instagram, father-related websites, seven primary schools, and acquaintances of the researchers. The minimum sample size was estimated using G-power 3.1 [32]. The differences between the intervention and control groups were analysed using paired data. The estimated sample size was 68 participants (34 per group), with an effect size of 0.7, an α of 0.05, and a power of 0.8. This sample size was chosen based on a study by Morgan [17]. The free LINE Corporation application was used during the pre–post questionnaire survey, which could be accessed by all participants. Data were collected between 3 May 2022 and 3 September 2022.

This study was conducted in accordance with the principles of the Declaration of Helsinki. In addition, the following points were considered in accordance with the ethical principles of medical research involving human participants: (1) ethical standards were followed to promote and ensure consideration of all participants and protect their health and rights; (2) the participants were completely informed about all aspects of the research, including the objectives, methods, sources of funding, potential conflicts of interest, institutional affiliations of the researchers, expected benefits, and anticipated risks; and (3) the participants were informed of their rights to refuse participation or withdraw their consent to participate at any time without prejudice. After obtaining informed consent, all participants were provided with comprehensive information about the web-based programme before commencement. The checking of the consent box on the questionnaire signified the participants’ agreement to participate in this study. This study was approved by the Ethical Review Committee of the Faculty of Medicine of Tokyo Medical and Dental University (M2021-274).

### 2.3. Intervention

Our programme included six videos, and a handbook was distributed to reinforce the video content. Table 1 presents the details of the programme. The videos and handbook consisted of four types of content: healthy lifestyle behaviour, lifestyle-related diseases, healthy diet, and healthy physical activity; all contents were intended for fathers and children such that they could watch the videos together. The educational content was based on the theoretical framework of Bandura’s Social Cognitive Theory [33], according to which health behaviour can be changed by changing cognitive, behavioural, and environmental factors to prevent disease. The programme’s content was based on the components of this theory, such as self-efficacy, outcome expectations, knowledge, social support, and behavioural intentions. The videos were uploaded on YouTube in advance and sent to the participants’ smartphone applications every other Saturday at 9 am. They were designed to affect participants’ cognitive and behavioural factors. Each video consisted of an objective at the beginning; a quiz on the content; an easy-to-follow strategy for integrating healthcare, healthy diet, and physical activity into daily life; and a summary of the content. The videos were approximately 3 min long. After watching the video, a quiz about the content was provided to assess the participants’ levels of understanding. The handbook was designed to influence each participant’s cognitive and environmental factors. It consisted of four chapters, with a checklist at the beginning of each chapter to motivate the fathers and children to reflect on their daily lives, diets, and physical activity. In addition, current issues related to the fathers’ and children’s lifestyles and the benefits of healthy behaviours were presented in order to motivate them. In addition, the handbook was designed to be enjoyable for the fathers and children to read and follow, including a column of stickers on each page requiring the children to paste stickers onto the pages read by the fathers. When half of the programme was completed, individual messages were sent to the participants, providing the information they wanted and offering some feedback on the programme. The control group was provided with a video and a handbook after the end of the study. The programme commenced on 3 May 2022 and ended on 10 December. The programme was thoroughly vetted and assessed for validity and reliability in our laboratory.

### 2.4. Measures

#### 2.4.1. Demographic Variables

All demographic variables and all outcome measures were collected via an online questionnaire. Data on the participants’ ages, family types (nuclear family, extended family, or others), and educational background (high school, vocational school, university, or graduate school); underlying diseases (no problem, obesity, hypertension, hyperlipidaemia, diabetes, or others); and the age and sex of the children were collected. Underlying diseases were only collected for the father.

#### 2.4.2. Primary Outcome

The outcome measures included changes in the participants’ awareness and behaviours regarding diet, physical activity, and healthcare. Awareness and behaviour were assessed using a five-point ordinal scale ranging from 1 (I do not know/not at all) to 5 (I know/I always do); the recommended diet, physical activity, and healthcare were determined based on the Ministry of Health, Labour, and Welfare guidelines for the prevention of lifestyle-related diseases [34]. For assessment of the total daily physical activity, the Shorter Activity Questionnaire developed by Imai et al. was used [35]. At the beginning of the questionnaire, the following general question was provided: ‘How much time do you usually spend in physical activity in a day, including work?’. Below the general question, more specific questions and their corresponding response options were provided: ‘Do you perform any muscle work or strenuous sports?’ (none, <1 h, or >1 h), ‘How much time do you spend sitting?’ (<3 h, 3–8 h, or >8 h), and ‘How much time do you spend walking or standing?’ (<1 h, 1–3 h, or >3 h). Total daily energy consumption was calculated by subtracting the time spent on all three types of activities and ‘other types of activities’ from the 24 h period. In addition, we assessed the stages of change in dietary and physical activity behaviour using a more detailed seven-point ordinal scale based on the transtheoretical model [36]. The participants were assessed before (T0: May 2022), 3 months after (T1: August 2022), and 4 months after (T2: September 2022) the intervention. We conducted measurements again four months after the intervention to confirm the durability of the effect. In addition, this study did not measure ecological indicators such as BMI because it was conducted over a short period of three months and focused on changes in fathers’ behaviour and knowledge.

#### 2.4.3. Secondary Outcomes

The participants’ awareness and management of their children’s diets, physical activity, healthcare, and changes in their children’s behaviour were rated on a five-point ordinal scale ranging from 1 (I do not know/not at all) to 5 (I know/I always do). The participants were assessed at the same time as the primary outcome measures. In cases where there were school-age siblings, respondents were asked to provide information on the lifestyle of the eldest child. This is because it has been reported that lifestyles tend to become worse with advancing age [37].

#### 2.4.4. Process Evaluation

The items programme duration, programme structure, timing of video delivery, length of video, amount of information in the handbook, change in health awareness, changes in awareness and behaviour of children and spouses, and satisfaction with the programme were rated on a five-point ordinal scale from 1 (too bad/no change at all) to 5 (very good/very good change). The process evaluation was completed by fathers using an online questionnaire 3 months after the intervention.

### 2.5. Analysis

As the continuous variables (father’s and children’s age, total daily physical activity) showed non-normal distribution (*p* < 0.05), a non-parametric test, the Mann-Whitney U test, was performed. As the other variables were expressed as nominal and ordinal scales, non-parametric tests were performed. Friedman tests were used to perform the pre–post comparisons of the outcomes for each group. The Fisher’s exact test and Mann–Whitney U test were performed to compare the differences in characteristics between the groups. The effect sizes (r) were calculated in each analysis using Cohen’s method and interpreted as small (r = 0.1), medium (r = 0.3), or large (r = 0.5) [38]. Fisher’s direct method was used to assess the relationship between the presence or absence of changes in health awareness and other items. The number of video views was divided into two groups based on the median: a low-level-of-video-viewing group and a high-level-of-video-viewing group. In addition, an intention-to-treat analysis was carried out to adjust for the number of dropouts. All data were analysed using SPSS Statistics (IBM, Armonk, NY, USA) ver. 23 for Windows. Statistical significance was set at a *p*-value of <0.05.

## 3. Results

### 3.1. Baseline Data

Figure 1 depicts a flowchart of the participant selection process. We enrolled 99 participants in the programme; however, 10 did not consent to participate and were not randomised. Therefore, 89 participants were allocated randomly to the different groups, with 46 and 43 participants in the intervention and control groups, respectively. Of them, 35 (76.1%) and 38 (88.4%) participants in the intervention and control groups, respectively, were included in the analysis after excluding those without follow-up outcome data (11 [23.9%] in the intervention group and 5 [11.6%] in the control group). Table 2 presents a summary of the participants’ characteristics. No significant differences were observed in the baseline demographic and clinical characteristics between the two groups. The average age of the participants’ children was 9.1 ± 1.6 years in the intervention group and 9.1 ± 1.7 years in the control group, with no significant differences between the two groups. Additionally, no significant differences were observed in the characteristics between those who continued to participate in the study until the 4-month follow-up and those who withdrew from the study.

### 3.2. Changes in the Primary Outcomes

Table 3 summarises the changes in the primary outcomes. The intervention group demonstrated significant improvements in the four aspects of dietary awareness: ‘a balanced diet’, ‘the recommended daily intake of energy’, ‘the guidelines for adequate daily vegetable intake’, and ‘the recommended amount of snacks per day’ (all *p* < 0.001). In terms of dietary behaviour, the intervention group demonstrated significant improvements in the following five items: ‘number of vegetable dishes per day’, ‘I choose my meals keeping energy in mind’, ‘I try to avoid fatty foods as much as possible’, ‘I try not to eat too much salt’, and ‘I consider energy and time of day when snacking’ (all *p* < 0.05). In the between-group comparisons, T0–T1 represents the difference between baseline and the period immediately after the intervention, while T0–T2 represents baseline and one month after the end of the intervention. The following items were significantly improved in the intervention group: ‘I know about a balanced diet’ (T0–T1: *p* = 0.016, r = 0.29; T0–T2: *p* = 0.010, r = 0.31), ‘I know the recommended daily intake of energy’ (T0–T1: *p* = 0.029, r = 0.26), ‘I know the guidelines for adequate daily vegetable intake’ (T0–T1: *p* = 0.007, r = 0.32; T0–T2: *p* < 0.001, r = 0.42), and ‘I know the recommended amount of snacks per day’ (T0–T1: *p* = 0.007, r = 0.32; T0–T2: *p* = 0.019, r = 0.28). Moreover, the dietary behaviours of the intervention group significantly improved in terms of ‘number of vegetable dishes per day’ (T0–T1: *p* = 0.005, r = 0.34) and ‘I consider energy and time of day when snacking’ (T0–T1: *p* = 0.025, r = 0.27; T0–T2: *p* = 0.023, r = 0.27) compared to those in the control group.

In terms of physical activity awareness, the intervention group demonstrated significant improvements in the items ‘awareness of improved physical activity habits’ and ‘I know the recommended amount of physical activity’ (all *p* < 0.001). In terms of behaviour, the intervention group demonstrated significant improvements in ‘total daily physical activity’ (*p* < 0.001). The intervention group showed significant improvements compared to the control group in the items ‘awareness of improved physical activity habits’ (T0–T2: *p* = 0.041, r = 0.24) and ‘I know the recommended amount of physical activity’ (T0–T1: *p* = 0.002, r = 0.36; T0–T2: *p* = 0.010, r = 0.31). Meanwhile, no significant differences were observed in ‘total daily physical activity’ between the two groups.

In regard to the healthcare-related items, the intervention group demonstrated significant improvements in the ‘I know my correct weight’ (*p* = 0.034) and ‘I know my weight management guidelines and methods’ (*p* < 0.001) items. The intervention group demonstrated a significant improvement in the ‘knowing the guidelines and methods of weight management’ (T0–T1: *p* = 0.025, r = 0.27; T0–T2: *p* = 0.007, r = 0.32) item compared with the control group.

### 3.3. Changes in the Secondary Outcomes

Table 4 summarises the changes in the secondary outcomes. The intervention group demonstrated significantly improved interest in their children’s diets and physical activity (*p* = 0.007 and *p* = 0.006, respectively). In terms of children’s awareness of a healthy lifestyle (diet, physical activity, and screen time), the intervention group demonstrated a significant improvement in the ‘I know the recommended daily amount of physical activity’ (T0–T1: *p* < 0.001, r = 0.46; T0–T2: *p* < 0.001, r = 0.44) and ‘I know my child’s correct weight’ (T0–T1: *p* = 0.001, r = 0.40, T0–T2: *p* = 0.001, r = 0.38) items compared with the control group. However, there were no significant intervention effects between the groups on behaviour change in children’s diet, physical activity, or screen time.

### 3.4. Process Evaluation

Figure 2 presents the process evaluation results. Approximately 76.1% and 88.4% of the participants in the intervention and control groups, respectively, were retained at up to 4 months of follow-up; 18 (52.9%) participants in the intervention group were satisfied with the programme, 20 (58.8%) experienced a change in the level of health awareness, and 10 (29.4%) reported a positive impact of the programme on their families. Sixteen (47.1%) participants watched all the videos. Table 5 shows the relationship between changes in health awareness and other items. Based on the median number of times that the videos were watched, the participants who watched the videos <4.5 times were classified as the low-level-of-video-watching group, while those who watched them >4.5 times were classified as the high-level-of-video-watching group. Changes in health awareness were correlated with a high level of video watching (*p* = 0.013) and a positive impact on family members (*p* = 0.002).

## 4. Discussion

### 4.1. Changes in Dietary Awareness and Behaviour

The intervention group showed a significant improvement in their level of dietary awareness after the intervention compared to the control group. Regarding dietary behaviour, there was a significant improvement in ‘the number of vegetable dishes consumed per day’. In Japan, the target daily vegetable intake for adults is ≥350 g (five or more small plates), and eating 350 g of vegetables per day can significantly reduce the burden of cardiovascular disease, cancer, and diabetic nephropathy [39]. By promoting awareness of a healthy diet and increasing vegetable intake among the participants, this programme may prevent possible future lifestyle-related diseases as well as maintain and improve the current health of the participants. In addition, the children in families wherein parents adhere to a balanced diet involving a high intake of vegetables tend to have similar dietary habits [40]. Moreover, parents influence their children’s food perceptions and choices [41]. Thus, vegetable intake among fathers may lead to an increase in their children’s vegetable intake. Dietary awareness and behaviour can be influenced by the person preparing the meal. However, the programme was designed to develop the ability of the subjects themselves to adjust their diets, for example, to choose an appropriate meal for lunch at work and to consider the time of day when they eat. Therefore, the changes in the awareness and behaviour of the intervention group brought about by the programme are considered meaningful. The intervention group demonstrated a significant improvement in the item ‘I consider energy and time of day when snacking’ compared with the control group. Individuals who engage in time-of-day-conscious eating are more likely to experience reductions in body weight and waist circumference compared with those who do not [42]. In the future, it will be necessary to study the time of day and the foods available to the participants and to adapt the programme to their living environments so that energy and time of day can be taken into account not only for snacking but also for food intake.

### 4.2. Changes in Physical Activity Awareness and Behaviour

The participants in the intervention group demonstrated significant improvement in their ‘awareness of improving physical activity habits’ and the ‘recommended amount of physical activity’. The programme focused on living activity rather than physical activity, with an emphasis on encouraging walking to work, using stairs, and reducing sitting time, aspects that can easily be incorporated into daily life. Through the programme, the participants reflected on their own physical activity habits and increased their awareness of familiar physical activities and the time spent on them, which improved their previous habits. In addition, the ‘total daily physical activity’ level significantly increased. To support behaviour change, experts should identify an individual’s current stage of behaviour change and problems, provide information on the benefits of action, and increase awareness and motivation [43]. Hence, the changes in the participants’ awareness of physical activity were evidenced by the increase in the amount of physical activity. However, physical activity also increased significantly in the control group. Therefore, no significant differences were found between the two groups before and after the intervention. Japanese men in their 40s and 50s have the lowest rate of habitual physical exercise among all other generations [3] owing to their ‘lack of time’ to engage in this activity [3]. In this context, the control group may have been motivated to be physically active based on their responses to the questionnaire. Liliana et al. [44] reported that self-monitoring and behavioural feedback are effective in promoting physical activity, and an intervention using an application or tracker had a positive effect, inspiring the participants to take 1850 steps per day. In order to promote physical activity among participants in the future, it would be useful to have an application that allows the participants to record and visualise their daily activities and parameters that allow them to feel the effects (weight, waist circumference, the number of steps, and blood test results). It is then important to provide regular feedback to the subjects to motivate them to be more physically active and to put this into practice.

### 4.3. Changes in Healthcare Awareness and Behaviour

The intervention group demonstrated a significant improvement in ‘knowing weight management guidelines and methods’ compared with the control group. Conversely, no changes were observed in the ‘checking their weight regularly’ item before and after the intervention. The reasons for not actually weighing oneself could be lack of time, lack of feeling of weight loss, low self-esteem due to weighing [45], and not having a scale. Daily weighing is generally associated with weight loss, while not weighing for >1 week, especially for >1 month, is associated with weight gain [46]. In addition to the daily monitoring of health status, regular weight management is expected to cause weight loss. Therefore, researchers should understand participants’ awareness of health management and provide recommendations tailored to their needs and lifestyles.

### 4.4. Changes in Children’s Interest in Lifestyle

The intervention group demonstrated significantly improved interest in their children’s diets and exercise habits. Moreover, the intervention group improved significantly in terms of their awareness of the recommended duration for their children to perform physical activity compared with the control group. The children of parents who are more interested in their lifestyles have healthier lifestyles than those of parents who are less interested [5]. Therefore, it is very important for parents of school-age children, whose lifestyle habits are still being established, to take an interest in their children’s lifestyle habits in order for them to grow and develop in a healthy way. However, no change was observed in the fathers’ behaviour in regard to managing their children’s diets. This finding may be attributed to the central role of mothers in managing their children’s diets [14]. It is not easy for children to regulate their own diets [47]. Therefore, in order for children to develop appropriate eating habits, it is important to educate both the father and the mother and to encourage the child from both sides. In addition, the intervention group demonstrated a significant improvement in their awareness of the ‘correct weights for children’ compared with the control group. The Japanese School Health Statistics Survey [48] stated that most children display a tendency to develop obesity at the age of 11 years, particularly boys, and this proportion exceeds 10% staring at the age of 9 years. Lifestyle and genetic factors have been implicated in the development of childhood obesity [49], and parental underestimation of a child’s actual weight is significantly associated with an increased childhood body mass index [50]. Therefore, it is important for parents to be acutely aware of their child’s weight in order to maintain the appropriate physique. If they have inaccurate perceptions, there is a need for health education that enables parents and children to reflect on their lives together with accurate information.

### 4.5. Process Evaluation

More than half of the participants in the intervention group were satisfied with the programme and experienced an improvement in their level of health awareness. Health-conscious fathers have better dietary habits [47], and higher parental health awareness influences children’s health habits [51]. Thus, improving the health awareness of fathers may exert a positive effect on their own habits as well as on those of their children. These findings support the association between changes in health awareness and their impact on families. In addition, 16 (47.1%) participants watched all the videos; the group that watched more videos demonstrated changes in health awareness, thus suggesting that the videos might have had an effective impact on fathers’ health awareness. The strength of web-based e-learning systems is that individual sessions can be conducted without a facilitator, allowing a programme to be completed at one’s own pace [29]. Therefore, the regular delivery of videos, in addition to the handbook, might have raised the awareness of the participants who were busy with their daily work.

### 4.6. Strengths and Limitations

The strengths of this study include its randomised controlled trial design, high retention rate, implementation of a programme based on theory, and use of social media. In addition, the fathers’ increased interest in their own health and their children’s lifestyles may improve the health of the entire family. Providing health education through social media, even virtually, and timely information from health professionals is considered cost-effective for busy fathers. However, this study has three limitations. First, we encountered difficulties in recruiting participants, and the sample size was relatively small. Hence, future studies should use better strategies for recruiting participants and providing incentives. Our study might suffer from participant selection bias. While the university enrolment rate in Japan in recent years has been 57.7% [52], the fathers who participated were characterised by the fact that around 70% or more of them had a university degree. The fathers may also have been more concerned about their children’s health and education. Therefore, there are limitations to generalising the findings of this study. Second, the self-administered questionnaire produced a social desirability bias. In addition to this questionnaire, the use of biological indicators, such as body weight and blood data, would be more appropriate for confirming the changes in the participants’ level of awareness and behaviour. Furthermore, some of the questionnaire items were developed by the researchers and need to be validated in the future. Third, 97.1% of the participants watched the videos at least once, but only 47.1% watched all six. Further studies should use content that meet the interests and needs of the target audience and provide videos that are suitable for time-constrained target audiences.

## 5. Conclusions

Social-media-based diet and physical activity programmes for fathers improved their awareness and behaviour. Although no indirect effects on children were observed, it was suggested that this could lead to fathers becoming more interested in their children’s lifestyles and health care. Therefore, social-media-based health education programmes can be important tools for increasing fathers’ interest in their own health and their children’s lifestyles.

## Figures and Tables

**Figure 1 ijerph-21-01104-f001:**
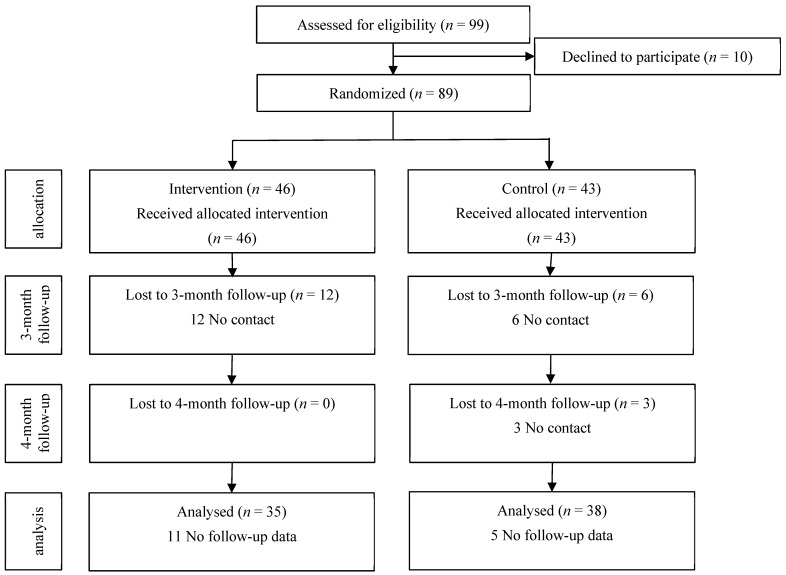
Trial flow chart of eligible participants in the intervention group and the control group.

**Figure 2 ijerph-21-01104-f002:**
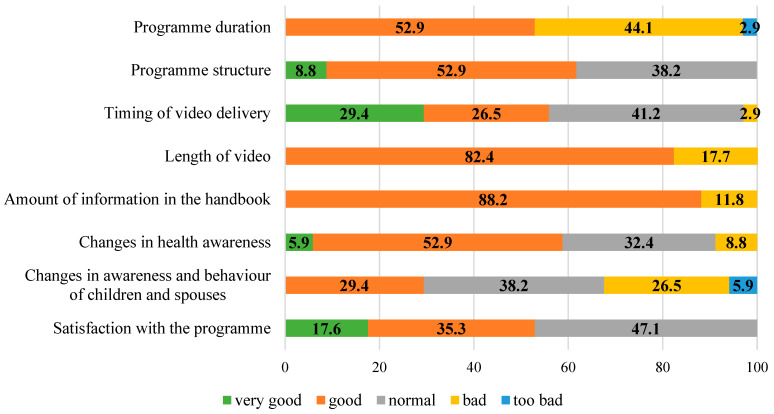
Evaluation of the programme.

**Table 1 ijerph-21-01104-t001:** Contents of the Healthy Lifestyles for Fathers and Children programme.

Content Focus	Tools	Content Detail	Component
Healthy lifestyle behaviour	Online video	Introducing the programme and researchers	Social support Knowledge
		Programme objectives	
		Quiz on lifestyle associations between fathers and children	
	Handbook	Factors influencing children’s lifestyles	Social support Knowledge Outcome expectations
		Importance of children’s lifestyles	
		Importance of reflecting on father-child lifestyles	
		Changes in fathers lead to changes in children	
Lifestyle-related diseases	Online video	Quiz on the benefits of weight loss	Behavioural intentions Knowledge Behavioural capability
		Weight control guidelines and methods	
		Monitor weight regularly	
		Health literacy (how to find the correct information)	
	Handbook	Review of father-child lifestyles	Social support Knowledge Outcome expectations Self-efficacy
		What are lifestyle diseases?	
		The difference between obesity and metabolic syndrome	
		How to manage weight	
		Children’s health issues	
		Children’s sleep and screen time	
Healthy diet	Online video	(1) Quiz on a balanced diet	Behavioural intentions Knowledge Behavioural capability
		Monitoring the contents of daily meals	
		Specific ways to achieve a balanced diet	
		(2) Quiz on seasonal vegetables	
		Monitoring the vegetable intake	
		Benefits of ‘eating with children’	
	Handbook	Review of the daily eating habits	Social support Knowledge Outcome expectations Self-efficacy Behavioural intentions
		Current dietary habits	
		What is a balanced diet?	
		Benefits of a balanced diet	
		Benefits of eating more vegetables	
		Recommended vegetable intake	
		Vegetable recipes to enjoy together	
		Daily energy requirements	
		Timing and choice of meals	
		Appropriate amount and timing of snacks	
Healthy physical activity	Online video	(1) Quiz on energy expenditure during physical activity	Behavioural intentions Knowledge Behavioural capability
		Key to increasing usual activity by 10 min	
		Begin with physical activity that matches the lifestyle	
		(2) Sedentary time quiz	
		Monitoring of sedentary time	
		Exercises for father and child	
	Handbook	Review of the daily activity level	Social support Knowledge Outcome expectations Self-efficacy Behavioural intentions
		Recommended levels of physical activity	
		Benefits of physical activity	
		What is physical activity?	
		What is Plus Ten?	
		Plus Ten to work on as a family	
		Physical activity for children’s development	
		The key to getting the child active	

**Table 2 ijerph-21-01104-t002:** Baseline characteristics of fathers.

Characteristics		Intervention (n = 46)		Control (n = 43)	
		Mean	SD	Mean	SD
Fathers age (years)		42.71	5.2	43.33	5.6
Family type, n (%)	Nuclear family	40	87	40	93
	Extended family	4	8.7	3	7.0
	Other	2	4.3	0	0.0
Educational background, n (%)	Highschool	5	10.9	5	11.6
Educational background, n (%)	Vocational school	5	10.9	7	16.3
Educational background, n (%)	University	27	58.7	24	55.8
	Graduate school	9	19.6	7	16.3
Underlying disease (allow multiple answers), n (%)	No problem	21	45.7	23	53.5
Underlying disease (allow multiple answers), n (%)	Obesity	10	21.7	13	30.2
Underlying disease (allow multiple answers), n (%)	Hypertension	7	15.2	7	16.3
Underlying disease (allow multiple answers), n (%)	Hyperlipidaemia	7	15.2	5	11.6
	Diabetes	1	2.2	1	2.3
	Other	6	13.0	5	11.6

SD, Standard deviation.

**Table 3 ijerph-21-01104-t003:** Changes in primary outcome variables from baseline to 3 and 4 months between the groups.

Intervention							Control											
N = 35							N = 38											
	T0		T1		T2			T0		T1		T2			T0–T1		T0–T2	
Variables	Med	[IQR]	Med	[IQR]	Med	[IQR]	*p* ^a^	Med	[IQR]	Med	[IQR]	Med	[IQR]	*p* ^a^	*p* ^b^	*r*	*p* ^b^	*r*
Behavioural change stages in diet ^c^																		
Awareness of improved eating habits	4.00	[2.00, 5.00]	5.00	[2.00, 5.00]	5.00	[3.00, 5.00]	0.399	3.00	[3.00, 5.25]	5.00	[2.50, 5.50]	4.00	[3.00, 5.00]	0.852	0.536		0.464	
Dietary awareness and behaviours ^d^																		
I know about a balanced diet	4.00	[2.00, 4.00]	4.00	[3.75, 4.00]	4.00	[4.00, 5.00]	0.000	4.00	[3.00, 4.00]	4.00	[3.00, 4.00]	4.00	[3.00, 4.00]	0.637	0.016	0.290	0.010	0.310
I know the recommended daily intake of energy	4.00	[2.00, 4.00]	4.00	[3.00, 4.00]	4.00	[4.00, 4.00]	0.000	4.00	[2.00, 4.00]	4.00	[2.50, 4.00]	4.00	[3.00, 4.00]	0.546	0.029	0.260	0.166	
I know the guidelines for adequate daily vegetable intake	2.00	[1.00, 4.00]	4.00	[3.00, 4.00]	4.00	[4.00, 4.00]	0.000	2.00	[2.00, 4.00]	3.00	[2.00, 4.00]	3.00	[2.00, 4.00]	0.047	0.007	0.320	0.000	0.420
I know the recommended amount of snacks per day	2.00	[1.00, 3.00]	3.00	[2.00, 4.00]	4.00	[2.00, 4.00]	0.000	2.00	[2.00, 3.00]	2.00	[1.00, 4.00]	3.00	[2.00, 4.00]	0.006	0.007	0.320	0.019	0.280
I eat at least two meals a day, including a main course, a main dish and a side dish	4.00	[2.00, 5.00]	4.00	[3.00, 5.00]	4.00	[4.00, 5.00]	0.114	4.00	[2.75, 5.00]	4.00	[2.50, 5.00]	4.00	[4.00, 5.00]	0.559	0.317		0.380	
Number of vegetable dishes per day	2.00	[1.00, 3.00]	3.00	[2.00, 4.00]	3.00	[2.00, 4.00]	0.001	2.00	[1.00, 3.00]	2.00	[1.00, 2.50]	2.00	[1.00, 3.00]	0.603	0.005	0.340	0.076	
I choose my meals with energy in mind	3.00	[2.00, 4.00]	4.00	[2.00, 4.00]	4.00	[2.00, 5.00]	0.015	3.00	[2.00, 4.00]	4.00	[2.00, 4.00]	4.00	[3.00, 4.00]	0.017	0.922		0.956	
I eat breakfast every day	5.00	[4.00, 5.00]	5.00	[4.00, 5.00]	5.00	[4.00, 5.00]	0.468	5.00	[4.00, 5.00]	5.00	[4.00, 5.00]	5.00	[4.00, 5.00]	0.125	0.195		0.203	
I try to avoid fatty foods as much as possible	2.00	[2.00, 4.00]	4.00	[2.75, 4.00]	4.00	[2.00, 4.00]	0.001	3.00	[2.00, 4.00]	3.00	[2.00, 4.00]	4.00	[3.00, 4.00]	0.045	0.156		0.370	
I try not to eat too much salt	3.00	[2.00, 4.00]	4.00	[3.00, 4.00]	4.00	[3.00, 4.00]	0.000	3.00	[2.00, 4.00]	4.00	[2.00, 4.00]	4.00	[3.00, 4.00]	0.042	0.187		0.682	
I consider energy and time of day when snacking	2.00	[1.00, 3.00]	4.00	[2.00, 4.25]	4.00	[2.00, 4.00]	0.000	2.50	[2.00, 4.00]	3.00	[2.00, 4.00]	4.00	[2.00, 4.00]	0.216	0.025	0.270	0.023	0.270
Behavioural change stages in physical activity ^c^																		
Awareness of improved physical activity habits	3.00	[2.00, 5.00]	4.50	[3.00, 6.00]	5.00	[3.00, 5.00]	0.000	3.00	[3.00, 5.25]	4.00	[3.00, 6.00]	4.00	[3.00, 6.00]	0.413	0.082		0.041	0.240
Physical activity awareness and behaviours ^d^																		
I know the recommended amount of physical activity	2.00	[1.00, 4.00]	4.00	[2.00, 4.00]	4.00	[3.00, 4.00]	0.000	2.50	[2.00, 4.00]	3.00	[2.00, 4.00]	3.00	[2.00, 4.00]	0.079	0.002	0.360	0.010	0.310
Total daily physical activity (METs)	36.50	[35.0, 37.3]	36.13	[34.6, 36.7]	38.50	[37.0, 41.8]	0.000	35.75	[33.3, 37.3]	35.75	[33.3, 36.5]	38.50	[37.8, 40.3]	0.000	0.627		0.771	
Healthcare awareness and behaviours ^d^																		
I know my correct weight	4.00	[4.00, 5.00]	5.00	[4.00, 5.00]	5.00	[4.00, 5.00]	0.034	5.00	[4.00, 5.00]	5.00	[4.00, 5.00]	4.00	[4.00, 5.00]	0.982	0.149		0.101	
I know my current BMI	4.00	[4.00, 5.00]	5.00	[4.00, 5.00]	5.00	[4.00, 5.00]	0.113	4.00	[3.00, 5.00]	5.00	[2.00, 5.00]	4.00	[3.00, 5.00]	0.284	0.123		0.385	
I know my weight management guidelines and methods	3.00	[2.00, 4.00]	4.00	[3.00, 5.00]	4.00	[4.00, 5.00]	0.000	4.00	[2.00, 4.00]	4.00	[2.50, 5.00]	4.00	[3.00, 4.00]	0.245	0.025	0.270	0.007	0.320
I check my weight regularly	4.00	[4.00, 4.00]	4.00	[3.75, 4.25]	4.00	[4.00, 5.00]	0.219	4.00	[2.00, 4.00]	4.00	[3.50, 5.00]	4.00	[3.00, 5.00]	0.358	0.190		0.784	

T0: before the intervention, T1: 3 months after baseline, T2: 4 months after baseline, BMI: body mass index; ^a^ Friedman test; ^b^ Mann-Whitney U test; ^c^ Seven-point ordinal scale, 1 = not interested in improving, 2 = interested but not intending to improve, 3 = intending to improve (within 6 months), 4 = intending to improve in the near future (within 1 month), 5 = already working on improvement (<6 months), 6 = already working on improvement (>6 months), 7 = other; ^d^ Five-point ordinal scale, 1 = do not know/not at all, 2 = do not know much/rarely, 3 = not sure, 4 = know quite a lot/sometimes, 5 = know/always.

**Table 4 ijerph-21-01104-t004:** Changes in secondary outcome variables from baseline to 3 and 4 months between the groups.

	Intervention						Control											
	N = 35						N = 38											
	T0		T1		T2			T0		T1		T2			T0–T1		T0–T2	
Variables	Med	[IQR]	Med	[IQR]	Med	[IQR]	*p* ^a^	Med	[IQR]	Med	[IQR]	Med	[IQR]	*p* ^a^	*p* ^b^	*r*	*p* ^b^	*r*
Children’s diet																		
Interest in my child’s diets ^c^	4.00	[4.00, 5.00]	4.00	[4.00, 5.00]	4.00	[4.00, 5.00]	0.007	4.00	[4.00, 5.00]	4.00	[4.00, 5.00]	4.00	[4.00, 5.00]	0.622	0.061		0.156	
My child eats breakfast every day ^d^	5.00	[5.00, 5.00]	5.00	[5.00, 5.00]	5.00	[5.00, 5.00]	0.819	5.00	[5.00, 5.00]	5.00	[5.00, 5.00]	5.00	[5.00, 5.00]	0.097	0.459		0.988	
I make sure that your child does noteat too many sweets ^d^	4.00	[2.00, 4.00]	4.00	[3.00, 5.00]	4.00	[3.00, 4.00]	0.223	4.00	[3.00, 4.00]	4.00	[2.00, 4.00]	4.00	[3.00, 4.00]	0.654	0.230		0.886	
I make sure your child does not eat too much oily food ^d^	3.00	[2.00, 4.00]	4.00	[3.00, 4.00]	4.00	[2.00, 4.00]	0.199	3.00	[2.00, 4.00]	4.00	[2.00, 4.00]	4.00	[2.00, 4.00]	0.294	0.658		0.787	
Children’s physical activity																		
Interest in my child’s exercise habits ^c^	4.00	[4.00, 5.00]	4.00	[4.00, 5.00]	5.00	[4.00, 5.00]	0.006	4.00	[4.00, 5.00]	4.00	[4.00, 5.00]	4.00	[4.00, 5.00]	0.395	0.254		0.125	
I know the recommended daily amount of physical activity ^d^	2.00	[1.00, 2.00]	3.00	[2.00, 4.00]	3.00	[2.00, 4.00]	0.000	2.00	[1.75, 3.00]	2.00	[2.00, 4.00]	3.00	[2.00, 4.00]	0.062	0.000	0.460	0.000	0.440
Frequency of physical activity outside of PE on weekdays ^e^	2.00	[2.00, 3.00]	2.50	[2.00, 3.00]	3.00	[2.00, 3.00]	0.984	2.00	[2.00, 3.00]	2.00	[2.00, 3.00]	2.00	[2.00, 3.00]	0.703	0.790		0.950	
Frequency of physical activity during holidays ^f^	3.00	[2.00, 4.00]	3.00	[2.00, 4.00]	3.00	[2.00, 4.00]	0.760	3.00	[2.00, 4.00]	3.00	[2.00, 4.00]	3.00	[2.00, 4.00]	0.747	0.380		0.800	
Children’s screen time																		
Time spent using computers and smartphones for purposes other than learning ^g^	2.00	[2.00, 3.00]	3.00	[1.75, 3.00]	3.00	[1.00, 4.00]	0.840	3.00	[2.00, 4.00]	3.00	[2.00, 4.00]	3.00	[2.00, 4.00]	0.759	0.970		0.820	
Child health care																		
I know my child’s correct weight ^c^	2.00	[1.00, 3.00]	3.00	[2.00, 4.00]	4.00	[2.00, 4.00]	0.000	2.00	[1.75, 4.00]	2.00	[1.00, 4.00]	3.00	[2.00, 4.00]	0.033	0.001	0.400	0.001	0.380

T0: before the intervention, T1: 3 months after baseline, T2: 4 months after baseline; ^a^ Friedman test; ^b^ Mann-Whitney U test; ^c^ Five-point ordinal scale, 1 = not interested, 2 = not very interested, 3 = undecided, 4 = quite interested, 5 = very interested; ^d^ Five-point ordinal scale, 1 = do not know/not at all, 2 = do not know much/rarely, 3 = not sure, 4 = know quite a lot/sometimes, 5 = know/always; ^e^ Four-point ordinal scale, 1 = never, 2 = one to two days a week, 3 = three to five days a week, 4 = every day; ^f^ Four-point ordinal scale, 1 = none, 2 = 1–3 days of monthly holidays, 3 = about half of the monthly holidays, 4 = every week; ^g^ Five-point ordinal scale, 1 = less than 30 min, 2 = more than 30 min but less than one hour, 3 = more than one hour but less than two hours, 4 = more than two hours, 5 = not applicable.

**Table 5 ijerph-21-01104-t005:** Correlations between health awareness, video viewing and family influence.

		Changes in Health Awareness				
		No		Yes		
		n	%	n	%	*p*-Value ^a^
Number of video views	a lot	3	21.4	14	70.0	0.013
	less	11	78.6	6	30.0	
Impact on the family members	yes	0	0.0	10	50.0	0.002
	no	14	100	10	50.0	

^a^ Fisher’s exact test.

## Data Availability

The datasets used and/or analysed in the current study shall be made available by the corresponding author upon reasonable request.

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
