# Peer review of "Effectiveness of a Social-Media-Based Diet and Physical Activity Programme for Fathers in Japan: A Randomised Controlled Trial"

_ijerph, 2024, doi:10.3390/ijerph21081104_

Round 1

Reviewer 1 Report

Comments and Suggestions for Authors

General comment:

It is an interesting study of an intervention focusing on fathers of primary school children in healthy lifestyles: diet, physical activity and healthcare. The aim was to clarify the effects of providing fathers with information on healthy lifestyle habits via social media on their awareness and behaviors regarding diet, physical activity, and healthcare, in addition to its indirect effects on their children. The intervention contents were intended for fathers and children and was delivered using social media principally available via smartphone applications using videos, and a handbook to reinforce the information of videos. The educative program content was based on the components of Social Cognitive Theory, and the change stages of Transtheoretical Model, a model that explains the process of behavior changes. The authors found that Social media-based diet and physical activity programs for fathers improved their awareness and behavior. They suggest that changes in father´s awareness and behavior can influence the lifestyle of their children.

Specific comments

1. The author should explain from what previous study or reason they estimated an effect size of .7, to obtain the sample size. The primary outcomes are many and what was used to calculate the sample size is not explained.

2. The times, which the measurements were performed (T0, T1 and T2) are not clearly written in their different sections of the manuscript. Apparently, the authors refer to the baseline measurement (T0), at the end of the intervention (T1) and at 1 month after the end of the intervention (T2). Furthermore, the reason for performing the last measurement is not mentioned.

3. In the introduction page 2, line 60 there is a mistake in the time/week in the Japanese men have paid work: “Japanese men have a long paid work time (452 min/week) [25]”

4. This sentence in inclusion criteria (page 3, lines 96-97) must be clarified: “who had access to a computer with internet facilities. Meanwhile, participants who did not own a smartphone were excluded.”

5. Some analyzes mentioned in the statistical analysis section were not later taken up in results or discussion sections. Is it necessary to maintain the description of those analyses for this manuscript?

6. In Table 2, a p value is omitted for the sex variable; in addition, 65.4% does not correspond to the 34/38 children in the control group. The characteristic of children age and sex could be mentioned in results only because table title is: Baseline characteristics of fathers. The baseline characteristics must be presented for the 43 and 46 participants that were allocated in each group, not only to those analyzed to primary outcome, and the p values are not necessary.

7. In table 2, what is the reason to divide the participants into the two groups in the variable of socio-economic status (people with financial comfort or not), how was the original variable measured? What does it mean, people with financial comfort or no? If the comparison between the groups is done to two categories variables, why is the variable with five categories presented in the table?

8. In table 3, why are mean and median presented to describe and compare variables? I suggested that only medians or n and % for ordinal variables be presented. Another option is to present the sum of the responses on the Likert scale of the different items included in the questionnaire and to make comparisons based on this score because there are many primary outcome variables. The same suggestion applies to the analysis of secondary outcome variables.

9. Figure 1 has problems of edition, the “0%” are in the wrong place. To improve the presentation of Figure 1, I suggest using colors instead of a range of whites and grays.

10. Please remove section 6. Patents.

Reviewer 2 Report

Comments and Suggestions for Authors

Dear Authors,

the paper is quite interesting and quite well written, however, in my opinion is too long and a bit repetitive in some parts.

In particular, you should try to summarize and synthesize the Material and Methods, Results (and tables), and Discussion sections as they appear hard to read.

Maybe you can also add a couple of paragraphs in the Conclusion section, which is very short.

Some specific comments are added in the attached file.

Best regards

Comments on the Quality of English Language

The English form is fine. A double-check with a mother language is always advisable.

Round 2

Reviewer 1 Report

Comments and Suggestions for Authors

Thanks to the authors for their responses and for incorporating the suggestions into this version.

I think that the statistical analysis could be improved if a score is obtained in each dimension of the questionnaire and not analyzed item by item.

Author Response

Comments1: I think that the statistical analysis could be improved if a score is obtained in each dimension of the questionnaire and not analyzed item by item.

Response1: Thank you for your very valuable comments. While we agree with you, there are two points of concern. First, as the scale used in this study is not an existing scale, we feel that it is not yet validated enough to give an overall score and compare results. This is also described in the limitations of the study (page.16, line.430-431). Second, we would also like to clarify which items affect the subjects, so that we can use this information as a consideration for future development of the scale. For this reason in this study we would like to present the results of our analysis for each item.